# Mechanical Properties, Crystallization Behaviors and Phase Morphologies of PLA/GTR Blends by Reactive Compatibilization

**DOI:** 10.3390/ma15207095

**Published:** 2022-10-12

**Authors:** Hongwang Shen, Yongxiang Hu, Zhitao Lin, Fantao Meng, Guannan Ju

**Affiliations:** School of Materials Science and Engineering, Shandong University of Technology, Zibo 255000, China

**Keywords:** polylactide, ground tire rubber, compatibilization, crystallization, balanced stiffness–toughness

## Abstract

Different ratios of Polylactic acid/Ground tire rubber (PLA/GTR) were prepared by melt blending and adding dicumyl peroxide (DCP) as a reactive compatibilizer. The compatibilizer could initiate a reaction between PLA and GTR to increase the compatibility and interfacial adhesion of the two phases, as indicated by Fourier transform infrared (FTIR) spectrometry and scanning electron microscopy (SEM). Adding the compatibilizer significantly improved the impact strength of the PLA/GTR blends without compromising the tensile strength. The elongation at the break and notched Izod impact strength of the blend increased by 61.8% and 150%, respectively, but there was only a 4.1% decline in tensile strength compared with the neat PLA. The plastic deformation on the impact fractured surface showed that the improvement of toughness could be attributed to the compatibilization initiated by DCP. Therefore, the improvement of the interfacial adhesion and compatibility of the two phases induced a brittle–ductile transition that occurred in the failure of blends. Moreover, the crystallinity of blends reached 40.5% without a further annealing treatment, which was nearly 24 times of the neat PLA, and the crystallization rate was enhanced simultaneously. These exciting findings suggest that compatibilization can provide a promising avenue for fabricating GTR-toughened PLA blends with balanced stiffness–toughness.

## 1. Introduction

Synthetic polymer materials have played an important role in daily life and the global economy. However, the production of more than 330 million tons globally every year requires huge consumption of petroleum resources and generates a large amount of polymer waste [1]. Suffering from some factors such as thermal, mechanical forces or UV irradiation, some polymer wastes are decomposed into microplastics [2], which can enter the human body through drinking water or food and then have detrimental effects on human health [3,4]. Hence, the replacement of traditional petroleum-based polymers by biodegradable and eco-friendly polymers is a task of urgent and high practical significance [5].

Among these polymers, polylactide (PLA) is one of the most essential representative polymers due to its outstanding mechanical properties, biocompatibility and biodegradability. Thus, the utilization of PLA may be considered as a promising alternative in order to replace conventional plastics for great potential application in various fields [6,7,8]. Nonetheless, some inherent drawbacks, especially brittleness, have restricted PLA from being used in extensive industrial applications [9]. Thereby, the enhancement in toughness is always the interesting point of the research and applications of PLA. Additionally, many strategies have been applied to improve PLA toughness by the copolymerization [10,11], plasticization [12,13], blending [9,14] and incorporation of the second composition [15,16]. Among various modification techniques, blending PLA with elastomeric polymer is considered to be the most effective way of improving the toughness, such as polycaprolactone (PCL) [17,18], polyhydroxyalkanoate (PHA) [19,20], poly(butylene adipate-co-terephtha-late) (PBAT) [11,21] and rubber [22,23]. For example, Chen et al. fabricated stable co-continuous PLA/PBAT blends, and the impact strength increased to 53.1 kJ/m^2^ [11]. However, the improvement in the impact strength always requires high elastomer content, which would lead to a drastic reduction in stiffness, such as the tensile strength decreasing by nearly 53% [24]. Recently, incorporating PLA and rubber via dynamic vulcanization can reduce the decline in tensile strength [14,23]. Nevertheless, the above elastomeric polymers are either fresh materials or rubber followed by vulcanization, which need to consume new resources and cannot lower the cost of PLA blends. Hence, it is hypothetical that some wasted elastomeric polymers, such as wasted vulcanized rubber, can be used to improve dramatically the toughness of PLA, and so this effective and economical strategy for obtaining an excellent stiffness–toughness balance is expected.

Ground tire rubber (GTR) can be obtained via different processes of waste tires to a desirable particle size [25]. Due to its low cost and good mechanical properties resulting from its cross-linked network structure, GTR has been incorporated in various composites such as thermoplastic [26], thermosets [27], concrete [28], rubber [29] and so on. The GTR in these composites has been found to improve impact strength and reduce crack propagation. However, many research works confirmed that the hydrophobic and cross-linked nature of GTR made it incompatible with plastic matrixes, which induced very poor mechanical properties of composite materials, therefore necessitating compatibilization to help improve its interaction adhesion with matrixes [30,31,32]. For instance, Sonnier et al. found that dicumyl peroxide (DCP) could enhance the compatibility of GTR and high-density polyethylene (HDPE), and then the better mechanical properties, elongation at break and impact energy of the composites were obtained [33]. Zedler et al. observed that the compatibility between GTR and acrylonitrile butadiene rubber (NBR) could be improved after the addition of an organic peroxide, which modified the mechanical and thermal properties of blends [34].

In the present study, DCP was used as a reactive compatibilizer to improve the structure and properties of PLA/GTR blends. Different ratios of PLA and GTR with or without DCP were evaluated in the reactive compatibilization, mechanical properties, crystallization behaviors and phase morphologies by FTIR, DSC, POM and SEM. From this work, we aimed to understand the effect of compatibilizer DCP on the miscibility and properties of PLA/GTR blends and find a promising avenue for fabricating GTR-toughened PLA blends with balanced stiffness–toughness.

## 2. Materials and Methods

### 2.1. Materials

Polylactide (PLA) (Ingeo 4032D) containing 2% _D_-lactide units, with an average molecular weight of 1.96 × 10^5^ g/mol and a dispersion index of 1.89 was purchased from Nature Works LLC, Minnetonka, MN, USA. GTR was obtained by the ambient grinding of wasted tire rubber with particles size of 80 mesh. Dicumyl peroxide (DCP, purity ≥ 99.5%) was purchased from Chengdu West Asia Chemical Co, China, and dichlomethane (DCM, CH_2_Cl_2_) was purchased from Sinopharm Chemical Reagent Co., Ltd (Shanghai, China). and used as received.

### 2.2. Samples Preparation

The PLA pellet and GTR powder were previously dried in a vacuum oven at 60 °C for 12 h. All the samples were prepared using an internal mixer Haake Rheomix at 180 °C and 60 rpm for approximately 10 min. The PLA was first melted and then GTR powder was added after 2 min followed by introducing DCP after 4 min. Finally, the blending continued for 4 min. The weight ratio of PLA to GTR was fixed as 95/5, 90/10, 85/15 and 80/20, and the content of DCP varied between 0.2, 0.6 and 1.0 wt.% for PLA/GTR blends (100 wt.%). The PLA/GTR blends are denoted herein as PLA/GTRx-Dy, where x and y are the wt.% of GTR and DCP, respectively. For instance, PLA/GTR5-D0.6 represents the blend containing 5 wt.% GTR and 0.6 wt.% DCP. Additionally, PLA with DCPs of 0.2, 0.6 and 1.0 wt.%, respectively, was also prepared as the comparison samples. Neat PLA was also subjected to the same mixing treatment to have the same thermal history as the blends. The specimens for the tensile and Izod impact tests were fabricated by the injection molding method, the materials were dried at 60 °C for 5 h and then injection molded in a Haake mini-injector at a barrel temperature of 185 °C and the mold temperature was 40 °C.

### 2.3. Characterizations and Evaluations

Fourier transform infrared (FTIR) spectrometry (Nicolet 5700, Thermo Fisher Scientific, Waltham, MA, USA) was used to study the interactions between PLA and GTR in the presence of DCP at a resolution of 4 cm^−1^ and a scanning number of 32. The spectra were recorded in wavenumbers from 4000 to 600 cm^−1^. The attenuated total reflectance mode was used for all tests. All samples were oven-dried to eliminate the effects of residual moisture prior to the test. Notably, DCM was used to selectively remove any free PLA molecular chains in PLA/GTR blends with or without DCP thoroughly (about 72 h under ambient temperature) in a Soxhlet extraction apparatus. After drying in a vacuum oven at 40 °C for 72 h, the extracted GTR (denoted as E-GTR and E-GTR-D) for FTIR was obtained.

The tensile properties were measured by using an Instron 5567 universal testing machine (Instron Corporation, Haiweikang, UK) with a cross-head speed of 10 mm/min according to the international standard ISO 527-2. The notched Izod impact strength was conducted on an electronic impact testing machine (XJJD-5J, Chengde Jinjian, Chengde, China) according to the international standard ISO 179-1. Before testing, all samples were conditioned for 48 h at 23 °C and humidity of 50%, and the reported data are the average values of at least five independent specimens for each sample.

The crystallization and melting behaviors of the blends were characterized by a Differential Scanning Calorimetry (DSC) instrument (TA Q2000, TA Instruments, New Castle, DE, USA) in the nitrogen atmosphere. All samples (about 6 mg sealed in an aluminum pan) were first heated at 40 °C/min from 40 °C to 200 °C and kept at this temperature for 3 min to erase the thermal history. Then, the samples were cooled down to 40 °C at 10 °C/min and heated again at the same rate. Both the exothermic and endothermic curves were recorded. All tests were repeated at least three times to evaluate the accuracy of the data. Crystallinity (*X*_c_) was then calculated using the following formula [35].
(1)$$X_{C}=\frac{\Delta H_{m}-\Delta H_{CC}}{\Delta H_{m}^{0}\left(1-m\right)}\times 100\%$$
where Δ*H*_m_ is the melting enthalpy from the second thermal scan, Δ*H*_cc_ is the measured enthalpy of cold crystallization during heating, and Δ$H_{m}^{0}$ is the theoretical melting enthalpy of fully crystallized PLA given as 93 J/g [35]. *m* is the weight fraction of GTR in the blends.

The crystal morphology and spherulite growth of the neat PLA and blends were observed by using polarized optical microscopic (POM) (Olympus BX51, Olympus Corporation, Tokyo, Japan) with a hot stage. The neat PLA and blend specimens sandwiched between two cover glasses were heated to 200 °C for 3 min on a hot plate and were pressed to prepare thin film samples with the desired thickness, and then the specimens were rapidly cooled to a temperature of 130 °C. POM observations were performed during isothermal crystallization at different times at this temperature.

The morphologies of GTR, the extracted GTR obtained by extracting blends using DCM, cryo-fractured and impact fractured surfaces of blends were examined by scanning electron microscopy (SEM, FEI Company, Hillsboro, OR, USA) at an accelerating voltage of 10 kV. The cryo-fractured surface was obtained after the samples were immersed in liquid nitrogen for 20 min, and the impact fractured surface was obtained from the samples after the Izod impact test. All samples were sprayed with gold for SEM observation. Besides, sulfur mapping analyses to identify GTR particles in the blends were carried out through energy dispersive X-ray (EDX) spectroscopy.

## 3. Results

### 3.1. Interaction between PLA and GTR

As we know, the compatibility of two phases has a great influence on the mechanical properties and microstructures of blends. The reaction may happen during melt-blending in the presence of DCP, which could improve the compatibility between PLA and GTR. In order to confirm that, blends were extracted by DCM for a period of 72 h to remove the free PLA phase completely. Therefore, FTIR was employed to analyze PLA, GTR, extracted GTR from PLA/GTR (E-GTR) and PLA/GTR with DCP (E-GTR-D), and the results are shown in Figure 1. As can be seen, the sharp absorption peak around 1750 cm^−1^ is caused by the carbonyl stretching vibration of C=O in PLA [9]. Moreover, the insoluble residues of E-GTR after the DCM extraction of the PLA/GTR blend show almost parallel FTIR spectra to that of GTR, indicating that free PLA molecular chains have been completely extracted by DCM and there is no reaction between PLA and GTR. As expected, compared with the GTR and E-GTR, the absorption peak at 1750 cm^−1^ also occurs in the spectra of the E-GTR-D (DCM-extracted GTR from PLA/GTR blend with DCP), which suggests the chemical reaction between PLA and GTR during melt-blending in the presence of DCP, because there is no free PLA in the insoluble residuals after extraction with DCM for 72 h.

FTIR spectra have preliminarily confirmed the reaction between PLA and GTR, which should be accompanied by changes in the surface morphology of GTR. The SEM was used to study these changes, further confirming the reaction between PLA and GTR, as shown in Figure 2. The surface of the GTR particle is very rough; there are many hollows and rugged places caused by application and grinding. Moreover, the surface morphology of the E-GTR particle is quite similar to that of the GTR particle, suggesting no reaction and the complete elimination of free PLA. However, in contrast to the GTR and E-GTR, the surface of the E-GTR-D particle is very smooth; it may be coated with bonded PLA due to the chemical reaction between these two phases, which further confirms the chemical reaction between PLA and GTR during melt-blending in the presence of DCP. This is consistent with FTIR results.

Based on the analysis above, the possible interfacial reaction between PLA and GTR in the presence of DCP is illustrated in Figure 3. The possible interfacial compatibilization mechanism at the PLA/GTR interface is promoted as follows. Both PLA and GTR can be initiated by the DCP radical-seizing hydrogen atom and form massive macromolecular free radicals. These two macromolecular free radicals interact with each other, and PLA molecules can be grafted onto the GTR to form grafted polymers (PLA-g-GTR) which act as the interfacial transition layer at PLA/GTR interface. Consequently, the compatibility and interface adhesion between the two phases can be improved, which is conducive to improving the mechanical properties of PLA/GTR blends.

### 3.2. Mechanical Properties

The effects of GTR content on mechanical properties are investigated for PLA/GTR blends with or without DCP, as shown in Figure 4. The tensile strength, elongation at break and impact strength of neat PLA are 63.3 MPa, 5.1% and 4.9 kJ/m^2^, respectively, in Figure 4a–c, reflecting a hard and brittle characteristic of PLA. All the samples without DCP show decreasing tensile strength as the content of GTR increases, as shown in Figure 4a. However, the elongation and impact strength show a similar trend in Figure 4b,c; these values both slightly increase at 5 wt.% GTR and then decrease with further increasing GTR content. Apparently, when blending PLA with GTR, not only can the toughness of PLA not be improved significantly but it also sacrifices tensile strength notably.

In the presence of DCP, though the changes in the mechanical properties of all samples have a similar trend to PLA/GTR blends without DCP, the tensile strength, elongation at break and impact strength all increase significantly after the addition of DCP. Interestingly, unlike PLA/GTR blends without DCP, with the incorporation of 5 wt.% GTR and 1.0 wt.% DCP, only a reduction of 4.1% in tensile strength is observed as compared with neat PLA, exhibiting a tensile strength of 60.7 MPa. Moreover, it can be seen that elongation at break and impact strength reaching to 12.3% and 14.3 kJ/m^2^, respectively, is observed with the GTR content of 5 wt.% with 1.0 wt.% DCP, which is about 2.5 and 3 times, respectively, that of neat PLA. In conclusion, the toughness of PLA can be significantly enhanced without much sacrifice in its stiffness after adding a certain amount of GTR and DCP simultaneously. This result can be contributed to the improvement of the compatibility of PLA and GTR, owing to the chemical reaction initiated by DCP. Moreover, it should be noted that the branching or crosslinking of the PLA matrix resulting from DCP might be a reason for such improvement. However, such a possibility is ruled out since the mechanical properties of neat PLA are not so sensitive to the DCP content. There are almost no changes in the tensile strength, elongation at break and impact strength of PLA with or without DCP in Figure 5. These results reveal that the remarkable improvement of mechanical properties is mainly attributed to the interfacial compatibilization between PLA and GTR in the presence of DCP.

Furthermore, the mechanical properties of the neat PLA and PLA/GTR blend (95/5) are investigated as a function of the DCP content, as shown in Figure 5. The tensile strength of PLA is increased by merely 1 MPa after the addition of DCP, and the effect of DCP on the elongation at break and impact strength is negligible. Notably, all the mechanical properties of PLA/GTR (95/5) increase with the increase in the DCP content, and its tensile strength, elongation at break and impact strength show maximum values of 60.7 MPa, 12.3%, and 14.3 kJ/m^2^ at 1.0 wt.% DCP, respectively. Especially, the elongation at break and impact strength increase by 61.8% and 150% of that of neat PLA, respectively. Therefore, DCP plays a very important role in the improvement of mechanical properties.

### 3.3. Crystallization Behavior

The non-isothermal crystallization behaviors of neat PLA and PLA/GTR blends with or without DCP are investigated by DSC, and the thermograms during programmed cooling and reheating processes are shown in Figure 6. The crystallization parameters derived from the DSC data are listed in Table 1, including the peak temperature of (cold) crystallization, *T*_c_ (*T*_cc_), the exothermic enthalpy of (cold) crystallization, Δ*H*_c_ (Δ*H*_cc_), the melting endothermic enthalpy (Δ*H*_m_), the difference between the onset crystallization temperature (*T*_onset_) and *T*_c_, the glass transition temperature (*T*_g_) and the melting temperature of crystals (*T*_m_).

As expected, neat PLA shows a flat curve. There are no (cold) crystallization peaks, and the endothermic peak of melting appears only as a slight bulge at about 166.4 °C, implying a low crystallization capacity, which is very similar to a result reported by Li et al. [36]. The addition of GTR brings up the distinct emergence of the crystallization peaks and the rapid shift of *T*_c_ to a low temperature in the cooling process with a minimum of 118.4 °C at 15 wt.% GTR. Additionally, then, during reheating, the cool crystallization peak is not observed, indicating the PLA has already crystallized to its maximum in the cooling cycle. However, the crystallization peak of PLA/GTR20 almost disappears, which appears only as a slight concave between 80 and 125 °C and is followed by a relatively weak cooling crystallization peak upon heating around 104.1 °C. Moreover, *T*_onset_ also decreases from 131.2 to 121.4 °C as an increase in the GTR content from 5 wt.% to 20 wt.%. These changes suggest that GTR may play a role in nucleating and significantly promoting the crystallization of PLA, but higher GTR content can decline this effect caused by uneven dispersion. Meanwhile, as a weathervane for the overall rate of crystallization, the smaller the value of *T*_onset_ -*T*_c_ is, the faster the crystallization completes. With the increase in the GTR content, the values decrease firstly and then increase with a minimum of 6.8 °C at 15 wt.% GTR addition. This effect further proves that the crystallization of PLA is effectively improved.

In an ordinary way, nucleating agents are mainly used to increase the crystallization rate, but are not necessarily helpful to the crystallinity at the same time. For all the samples with GTR, the crystallinities are significantly higher than that of neat PLA. The maximum *X*_c_ of 38.7% is observed at 15 wt.% GTR loading, which is about 23 times that of neat PLA. However, *T*_m_ shifts slightly to low temperature with an increase in the GTR content, indicating more imperfect crystallization. Thus, there is no doubt that GTR plays a significant role in nucleating and dramatically raises the crystallinity without a further annealing treatment.

The influence of DCP on the crystallization behaviors of PLA/GTR (95/5) also are observed. After the addition of 0.2 wt.% DCP, *T*_c_ and *T*_onset_ both decrease obviously but the value of *T*_onset_ − *T*_c_ reaches a maximum of 16.6. Meanwhile, the *X*_c_ decreases by 22.5%. Weak and broad crystallization peaks and cold crystallization peaks are observed, meaning that the low content of DCP depresses the crystallization of PLA/GTR. A further increase in the DCP content brings up a shift of *T*_c_ and *T*_onset_ to high temperature and a distinct decrease in the *T*_onset_ -*T*_c_ value with a strong and sharp crystallization peak. Notably, a dramatic increase in *X*_c_ (40.5%) is observed at 0.6 wt.% DCP, which suggests higher DCP content could dramatically improve the crystallinity and crystallization rate of PLA molecules This can be attributed to two reasons. The first one is the reduced crystallization induction period due to the presence of GTR (nucleating) and interfacial compatibilization caused by DCP; the second one is the increased molecular chain mobility associated with slight *T*_g_ depression [36], which accelerates the crystal growth. Thus the combination of the reduced *T*_g_ and the higher nucleation density results in the improvement of the crystallization rate and crystallinity simultaneously, which is very important for the PLA blends because the melt processing, such as injection molding or extrusion, needs a fast-molding time in industrial fields. This is seldom achieved by simple blending without additional nucleation agents and annealing treatment.

### 3.4. POM Observation

Polarized optical microscopy (POM) was adopted to monitor the crystal growth during the isothermal crystallization of PLA and PLA/GTR (95/5) blends with or without DCP (1.0 wt.%), as shown in Figure 7. It is clearly seen that the typical spherulite structure is observed in all samples, indicating that the addition of GTR and DCP does not alter the crystal form of PLA. However, the size and number of spherulites show a dramatic change. Compared with neat PLA, a smaller size and a higher quantity of crystals are observed in the PLA/GTR blend at the same time, and the spherulite interface is also very clear. It is noteworthy that PLA/GTR with 1.0 wt.% DCP presents a much higher nucleation density at the same time (i.e., 5 min), and thus the spherulite size decreases further, showing a large number of aggregates of tiny crystals, and the interface between the crystals becomes blurry. This result is consistent with the DSC analysis, indicating that GTR and DCP play a significant role in nucleating and dramatically raising the crystallinity. A large spherulite of small size would have a great effect on mechanical properties. 

### 3.5. Morphology Observation

The SEM images of cryofracture surfaces of neat PLA and PLA/GTR (95/5) with or without DCP (1.0 wt.%) are presented in Figure 8. As seen in Figure 8a, the fractured surface of neat PLA is very smooth with obvious brittle fracture characteristics, showing no plastic deformation. After incorporating with GTR, typical sea-island morphology is obtained for PLA/GTR blend, and the interfacial adhesion between PLA and GTR is very poor, as indicated by the void spaces throughout the fracture surface due to the GTR being pulled out of the PLA matrix during the cryofracture (Figure 8c). Adding DCP greatly improves the interfacial adhesion between PLA and GTR, with most GTR particles not debonding from the matrix, though some holes are still observed, which indicates superior compatibility between PLA and GTR. The apparent adhesion observed between PLA and GTR may be attributed to the occurrence of some chemical reactions between the two phases during melt-blending in the presence of DCP. Moreover, to further understand the dispersion and localization of GRT particles in the PLA matrix, energy dispersive X-ray (EDX) analyses were performed on PLA/GTR blends (Figure 8b), and the yellow spots in the EDX maps show the presence of sulfur. The EDX map shows a uniform distribution of sulfur, confirming an excellent dispersion of GTR particles. Similar morphologies are observed for the blends containing 10–20 wt.% GTR but with the agglomeration of GTR particles, which are not shown herein for the sake of brevity. In summary, DCP provides a refined micrograph with GTR particles being well-embedded and bonded within the PLA matrix. The enhancement of interfacial adhesion allows the blends to consume more energy when damaged and finally improves the impact toughness of the blends, which can explain why PLA/GTR blends with DCP show high impact strength.

The notched Izod impact strength of PLA/GTR blends is increased by 150% after the addition of 1 wt.% DCP (Figure 5c). This behavior is explained by the analysis of the impact fracture surfaces using SEM, as shown in Figure 9. Local magnification is shown in the upper right corner of the image. The impact cross-section of PLA/GTR is relatively smooth showing a brittle-like fracture (Figure 9a), as indicated by low impact strength similar to the neat PLA (Figure 5c). After the addition of DCP, the impact cross-section becomes rough, but a low DCP content (0.2 wt.%) does not obviously alter the morphology of the blends, and the GTR particles can be still observed clearly (Figure 9b). As the content of DCP increases, the GTR particles have not been already observed because of the more deformation of the PLA matrix. Interestingly, notable plastic deformation of PLA in blends with 1.0 wt.% DCP occurred during the impact process (Figure 9d). Thus, a brittle-to-tough transition of the blends is observed. This might be because the occurrence of some chemical reactions between PLA and GTR at high contents of DCP significantly increased the interfacial adhesion. When samples are subjected to impact force, the superior interfacial adhesion can transfer energy effectively to the GTR phase, thereby generating a large number of plastic deformation zones to absorb a large amount of energy and improve the impact strength of the blends.

## 4. Conclusions

In this paper, PLA-based materials with balanced stiffness–toughness were successfully prepared by melting reactive blending with GTR. The addition of DCP initiated free radicals in the blends, leading to grafted polymers such as PLA-g-GTR, which acted as a compatibilizer for PLA/GTR blends. As a consequence, the compatibility and interface adhesion between the two phases were improved obviously, which was evidenced by FITR and SEM. The mechanical properties of the PLA/GTR blends, notably the toughness without sacrificing tensile strength, were improved due to the compatibilization initiated by DCP. At a PLA/GTR ratio of 95/5 and the addition of 1.0 wt.% DCP, the elongation at break increased by 61.8%, and the notched Izod impact toughness increased by 150% but there was only 4.1% decline in tensile strength compared with the neat PLA. Moreover, GTR particles played a significant role in nucleating, and thus the crystallinity (nearly 24 times of neat PLA) and crystallization rate of PLA were dramatically enhanced simultaneously without a further annealing treatment after the addition of DCP.

## Figures and Tables

**Figure 1 materials-15-07095-f001:**
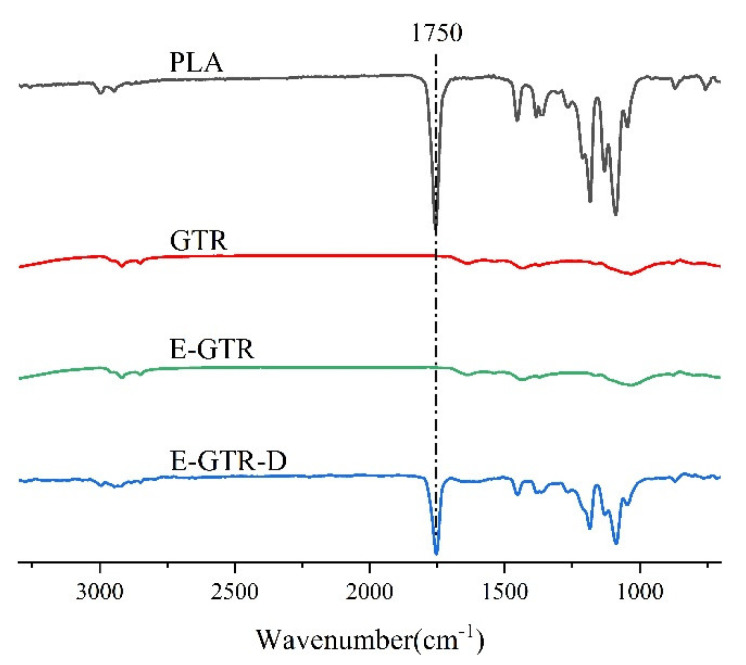
FTIR spectra of neat PLA, GTR, extracted GTR from PLA/GTR (E-GTR) and PLA/GTR with DCP (E-GTR-D).

**Figure 2 materials-15-07095-f002:**
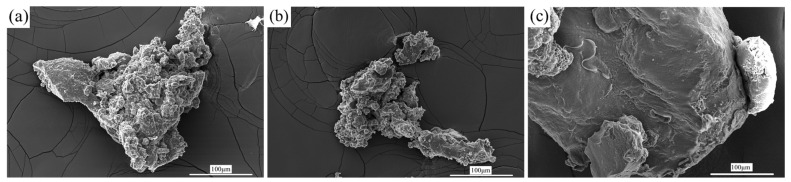
SEM images of (**a**) GTR. (**b**) Extracted GTR from PLA/GTR without DCP. (**c**) Extracted GTR from PLA/GTR with DCP.

**Figure 3 materials-15-07095-f003:**
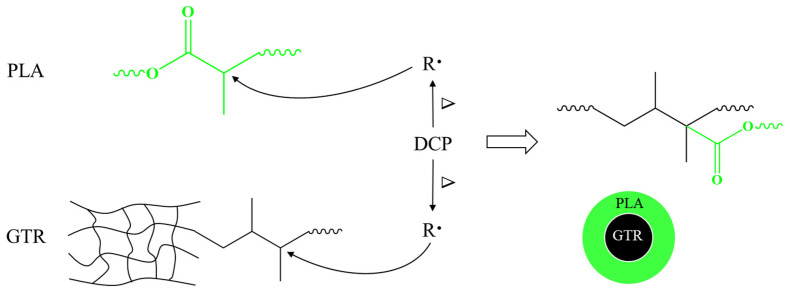
Schematic diagram of the possible interfacial reaction between PLA and GTR.

**Figure 4 materials-15-07095-f004:**
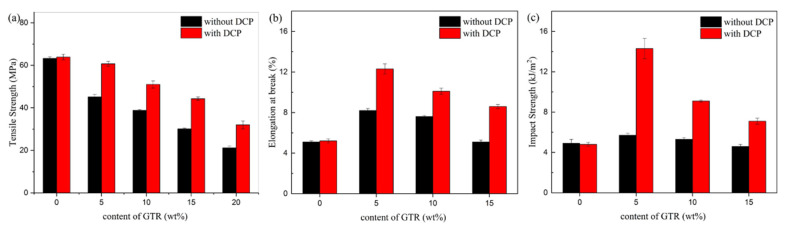
Mechanical properties of PLA as a function of GTR content with or without DCP: (**a**) Tensile strength. (**b**) Elongation at break. (**c**) Notched Izod impact strength.

**Figure 5 materials-15-07095-f005:**
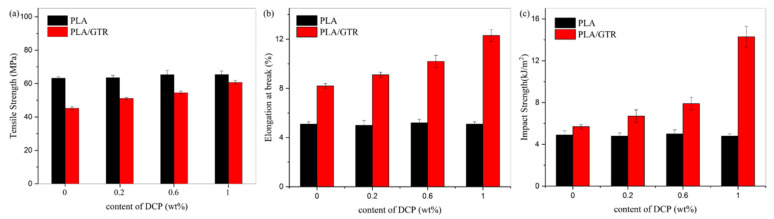
Mechanical properties of neat PLA and PLA/GTR (95/5) as a function of DCP content: (**a**) Tensile strength. (**b**) Elongation at break. (**c**) Notched Izod impact strength.

**Figure 6 materials-15-07095-f006:**
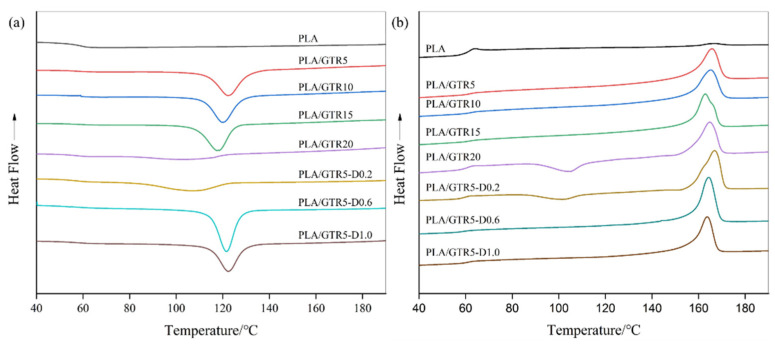
DSC curves of neat the PLA, PLA/GTR blends with or without DCP: (**a**) Crystallization. (**b**) Melting.

**Figure 7 materials-15-07095-f007:**
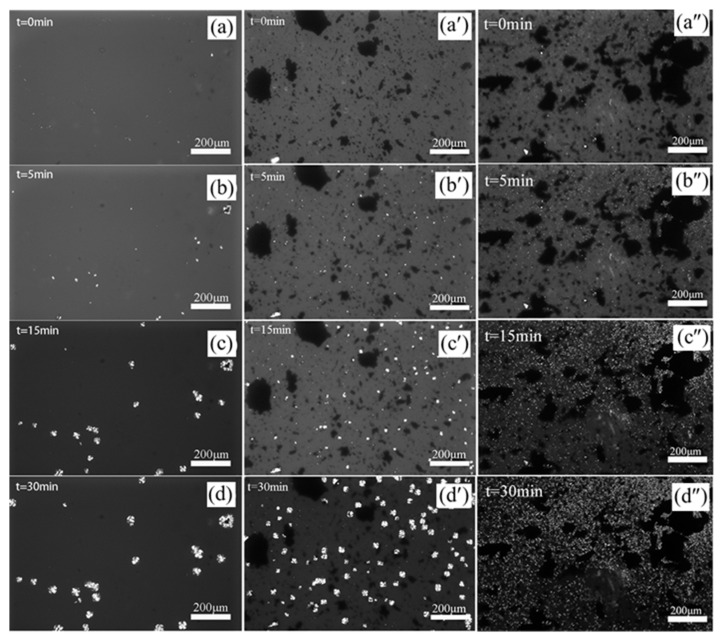
POM images of crystal growth at 130 °C with different times: (**a–d**) PLA. (**a′**–**d′**) PLA/GTR5. (**a″**–**d″**) PLA/GTR5-D1.0.

**Figure 8 materials-15-07095-f008:**
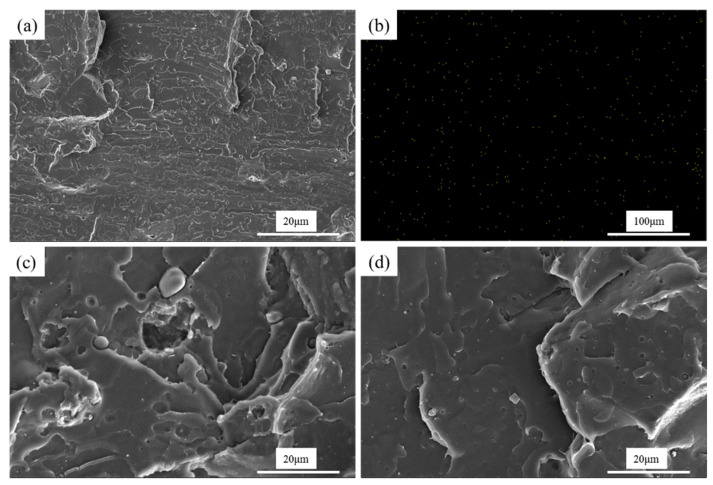
SEM images of cryofracture surfaces of (**a**) neat PLA. (**c**) PLA/GTR5. (**d**) PLA/GTR5-D1.0, and (**b**) EDX maps for PLA/GTR5.

**Figure 9 materials-15-07095-f009:**
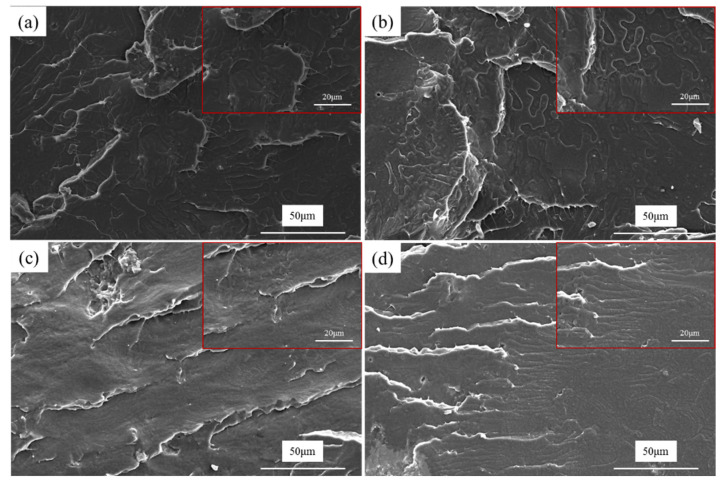
SEM micrographs of the impact fracture surfaces for PLA/GTR (95/5) blends with different DCP content: (**a**) 0 wt.%; (**b**) 0.2 wt.%; (**c**) 0.6 wt.%; (**d**) 1.0 wt.%.

**Table 1 materials-15-07095-t001:** Crystallization parameters of the PLA and PLA/GTR blends.

Samples	*T*_g_ (°C)	*T*_cc_ (°C)	*T*_c_ (°C)	*T*_onset_ (°C)	*T*_onset_ -*T*_c_ (°C)	*T*_m_ (°C)	∆*H*_cc_ (J/g)	∆*H*_c_ (J/g)	∆*H*_m_ (J/g)	*X*_c_ (%)
PLA	61.8	-	-	-	-	166.4	-	-	1.6	1.7
PLA/GTR5	61.2	-	122.4	131.2	8.8	165.8	-	33.4	32.2	36.4
PLA/GTR10	61.5	-	120.1	128.5	7.4	165.2	-	31.9	30.8	36.8
PLA/GTR15	61.5	-	118.4	125.2	6.8	163.1	-	31.6	30.6	38.7
PLA/GTR20	61.2	104.4	105.4	121.4	16.0	164.8	12.8	11.4	28.0	20.5
PLA/GTR5-D0.2	59.3	101.6	106.8	123.4	16.6	166.8	8.6	19.4	33.5	28.2
PLA/GTR5-D0.6	60.3	-	121.5	127.7	6.2	164.5	-	40.1	35.6	40.5
PLA/GTR5-D1.0	61.6	-	122.5	129.6	7.1	163.8	-	31.4	31.3	35.8

## Data Availability

Not applicable.

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
