# Peer review of "Mechanical Properties, Crystallization Behaviors and Phase Morphologies of PLA/GTR Blends by Reactive Compatibilization"

_materials, 2022, doi:10.3390/ma15207095_

Round 1

Reviewer 1 Report

This paper reports a method of adding DCP as a reactive compatibilizer into PLA to achieve balanced stiffness-toughness. The experimental design is sound and data can well support the conclusion. However, the English writing should be checked carefully and improved before publication. I found a lot of grammar mistakes in the manuscript. For example, in Line 10, "reactive" should be "reaction"; Line 52, "incorporation" should be "incorporating". There are more in the manuscript. 

Author Response

Point-by-point response to reviewer’ comments

We would like to thank the reviewers for taking their time to review our manuscript. Highlighted in yellow in the manuscript are the revisions that have been made.

Reviewer 1

This paper reports a method of adding DCP as a reactive compatibilizer into PLA to achieve balanced stiffness-toughness. The experimental design is sound and data can well support the conclusion. However, the English writing should be checked carefully and improved before publication. I found a lot of grammar mistakes in the manuscript. For example, in Line 10, "reactive" should be "reaction"; Line 52, "incorporation" should be "incorporating". There are more in the manuscript.

Reply: We have carefully checked and improved the English writing in the revised manuscript.

Reviewer 2 Report

The manuscript fall in the goal and objective of the journal. The author explained "Mechanical properties, crystallization behaviors and phase  morphologies of PLA/GTR blends with balanced stiffness- 3 toughness via reactive compatibilization". There are still some areas need to improve.

1. Title is not appropriate: suggestion: Mechanical properties, and phase  morphologies of PLA/GTR blends prepared by reactive compatibilization.

2. Abstract need to modified and need to checked by native english speaker.

3. 34 references has been added introduction, however, no references has been added to result and discussion to support their findings. 

Author Response

We would like to thank the reviewers for taking their time to review our manuscript. Highlighted in yellow in the manuscript are the revisions that have been made.

Reviewer 2

The manuscript fall in the goal and objective of the journal. The author explained "Mechanical properties, crystallization behaviors and phase morphologies of PLA/GTR blends with balanced stiffness- 3 toughness via reactive compatibilization". There are still some areas need to improve.

  1. Title is not appropriate: suggestion: Mechanical properties, and phase morphologies of PLA/GTR blends prepared by reactive compatibilization.

Reply: We have changed the tile into “Mechanical properties, crystallization behaviors and phase morphologies of PLA/GTR blends prepared by reactive compatibilization”.

  1. Abstract need to modify and need to check by native english speaker.

Reply: We have carefully checked and improved the English writing in the revised manuscript.

  1. 34 references has been added introduction, however, no references has been added to result and discussion to support their findings.

Reply: We have added some suitable references in the result and discussion part.

Reviewer 3 Report

The present paper demonstrates the compatibilizing effect of dicumyl peroxide (DCP) in blends of polylactide (PLA) and ground tire rubber (GTR). In general, the study is well performed but some points require attention before publication.

a)       Try to place Figures after the corresponding text (Figure 1, Figure 2, Figure 3, Figure 5, Figure 6 should be placed after being cited in the text).

b)      Change, please, the order of spectra in Figure 1: E-GTR should appear before E-GTR-D.

c)       Include, please, a scheme of chemical reactions that are described in the paragraph 181-188.

d)      Paragraph 213-219 should be improved since it is not clear.

e)      DSC curve of PLA in Figure 5b is strange since a cold crystallization is usually observed. Authors should improve discussion/justification or refer to published results.

f)        Table 1 should include the crystallization enthalpies. Note for example that the melting enthalpies of PLA/GTR20 and PLA/GTR5-D0.2 samples and clearly higher than the cold crystallization enthalpies while the corresponding previous crystallization curves are practically flat.

g)       Lines 279-288: The variation of the glass transition temperature is reduced (maximum around 2 ºC). Can this reduced change justify mobility differences?

h)      POM experiments: Can the authors indicated some numerical data (e.g., the nuclei density or even the estimation of spherulite diameters).

i)        Line 364: which is the evidence for the increase of the crystallization rate? I think that no experiments were performed to evaluate crystallization rates.

j)        English language requires a careful revision since mistakes are plentiful and explanation is in some cases deficient.

Author Response

Point-by-point response to reviewer’ comments

We would like to thank the reviewers for taking their time to review our manuscript. Highlighted in yellow in the manuscript are the revisions that have been made.

Reviewer 3

The present paper demonstrates the compatibilizing effect of dicumyl peroxide (DCP) in blends of polylactide (PLA) and ground tire rubber (GTR). In general, the study is well performed but some points require attention before publication.

  1. Try to place Figures after the corresponding text (Figure 1, Figure 2, Figure 3, Figure 5, Figure 6 should be placed after being cited in the text).

Reply: We prepared the manuscript according to MDPI template, and have placed these Figures after the corresponding text.

  1. Change, please, the order of spectra in Figure 1: E-GTR should appear before E-GTR-D.

Reply: We have changed the order of E-GTR and E-GTR-D, the E-GTR appears before E-GTR-D in Figure 1.

  1. Include, please, a scheme of chemical reactions that are described in the paragraph 181-188.

Reply: We have given a scheme of chemical reactions between PLA and GTR in the presence of DCP (Figure 3).

  1. Paragraph 213-219 should be improved since it is not clear.

Reply: There may be two reasons for the improvement of mechanical properties of blends, one is the interfacial compatibilization, and the other may be the molecular chains crosslinking of PLA initiated by DCP. But the second reason have been ruled out according to the result in Figure 5. We have modified the discussions about this part in the manuscript.

  1. DSC curve of PLA in Figure 5b is strange since a cold crystallization is usually observed. Authors should improve discussion/justification or refer to published results.

Reply: The DSC test was repeated three times, so it can guarantee data accuracy and authenticity. And the DSC curve of PLA use in this study is very similar to the result reported by Li et al [Refer 37]. We have improved this discussion in the manuscript by referring to this published paper.

  1. Table 1 should include the crystallization enthalpies. Note for example that the melting enthalpies of PLA/GTR20 and PLA/GTR5-D0.2 samples and clearly higher than the cold crystallization enthalpies while the corresponding previous crystallization curves are practically flat.

Reply: We have supplemented the data of crystallization enthalpies (∆Hc) in Table 1.

  1. Lines 279-288: The variation of the glass transition temperature is reduced (maximum around 2 ºC). Can this reduced change justify mobility differences?

Reply: The glass transition temperature is the temperature when the molecular chain segments begin to move, thus the decrease in the glass transition temperature can indicate that molecular chains can began to move at a lower temperature, that is, increasing the molecular chains mobility at given temperature. And this kind of explanation is also used in a published paper reported by Li et al [Refer 37].

  1. POM experiments: Can the authors indicated some numerical data (e.g., the nuclei density or even the estimation of spherulite diameters).

Reply: We have tried to estimate the nuclei density and the size of spherulite, but it is very difficult to observe because of the huge numbers of nuclei density and the undersized spherulite. And we think that the obvious changes in the POM images have indeed provided power evidences for the discussion about changes in crystallization behaviors.

  1. Line 364: which is the evidence for the increase of the crystallization rate? I think that no experiments were performed to evaluate crystallization rates.

Reply: Crystallization of polymers follows the theory of “nucleation first and then growth”, that is, molecular chains overcome a barrier and form crystal nuclei through molecular thermal motion and then the molecular chains grow into crystals around the center of the crystal nuclei. Thereby, the overall rate of crystallization is depend on the rates of these two processes. In DSC analysis, at the same cooling rate, there is a distinct decrease in Tonset -Tc value (a weathervane for the overall rate of crystallization) with a strong and sharp crystallization peak, and a dramatic increase in Xc (40.5 %) is observed at the same crystallization time, which is about 24 times than that of neat PLA. Besides, the POM images show a dramatically increase in the crystal nuclei density and a shorter crystallization time. These phenomenons are all the power evidences for the increase of the crystallization rate.

  1. English language requires a careful revision since mistakes are plentiful and explanation is in some cases deficient.

Reply: We have carefully checked and improved the English writing in the revised manuscript.

Reviewer 4 Report

This last version of manuscript is appropriate for publication. However. in the abstract some nouns of compounds and experimental methods are written by uppercase characters. Please, avoid them.

Author Response

Point-by-point response to reviewer’ comments

We would like to thank the reviewers for taking their time to review our manuscript. Highlighted in yellow in the manuscript are the revisions that have been made.

Reviewer 4

  1. This last version of manuscript is appropriate for publication. However, in the abstract some nouns of compounds and experimental methods are written by uppercase characters. Please, avoid them.

Reply: We have carefully checked and corrected the English writing in the abstract.

Round 2

Reviewer 3 Report

Authors have improved the manuscript. Most suggestions have been conveniently addressed, although some points are still weak. Nevertheless, I understand that no improvements can be performed at this stage.